# Decision-Making in Youth Team-Sports Players: A Systematic Review

**DOI:** 10.3390/ijerph17113803

**Published:** 2020-05-27

**Authors:** Ana Filipa Silva, Daniele Conte, Filipe Manuel Clemente

**Affiliations:** 1Escola Superior Desporto e Lazer, Instituto Politécnico de Viana do Castelo, Rua Escola Industrial e Comercial de Nun’Álvares, 4900-347 Viana do Castelo, Portugal; filipe.clemente5@gmail.com; 2N2i, Polytechnic Institute of Maia, 4475-690 Maia, Portugal; 3The Research Centre in Sports Sciences, Health Sciences and Human Development (CIDESD), 5001-801 Vila Real, Portugal; 4Institute of Sport Science and Innovations, Lithuanian Sports University, 44221 Kaunas, Lithuania; daniele.conte@lsu.lt; 5Instituto de Telecomunicações, Delegação da Covilhã, 1049-001 Lisboa, Portugal

**Keywords:** decision making, youth, sports, team sports, performance

## Abstract

The aim of this systematic review conducted in the topic of youth team-sports players was three-fold: (i) Analyze the variations of decision-making processes between low- and high-level youth players; (ii) analyze the variations of decision-making processes between different age groups; and (iii) analyze the effects of decision-making training-based programs on youth players. Following the preferred reporting items for systematic reviews and meta-analyses (PRISMA) guidelines, this systematic review searched for studies on PubMed, ScienceDirect, Academic Search Complete, SPORTDiscus, and Taylor & Francis Online. The search returned 6215 papers. After screening the records against set criteria, 26 articles were fully reviewed. From the included studies, 9 were focused on comparing the decision-making process between low- and high-level players, 6 compared the decisions made by players from different age categories, and 11 analyzed the effects of decision-making-based training programs on youth players. Comparisons between high- and low-level players suggested that high-level and most talented players present a greater accuracy in the cognitive and executive answers to the game as well as being more adjustable to more complex situations. Considering the comparisons between age groups, a tendency of older players to execute more accurate decisions in the game and to have better tactical knowledge and behavior was observed. Finally, the effects of decision-making training programs suggest a beneficial effect employing practical scenarios (mainly based on small-sided and conditioned games), primarily improving passing decisions and execution. However, the benefits of interventions using videos are not clear.

## 1. Introduction

Open-skill sports are characterized by the repetition of high-intensity actions that require athletes to possess well-developed physical and physiological characteristics, such as speed, strength, power, agility, and fitness [1,2]. Additionally, such sports require well-developed technical (e.g., passing, dribbling, and shooting) and tactical skills [3,4]. Specifically, team sports have been considered complex systems according to ecological dynamics theories [5]. Expressly, the interaction between players and the information given by the performance environment limit the occurrence of patterns of stability (i.e., coordination between performers), variability (loss of coordination between performers), and symmetry breaking in organizational states (i.e., how new patterns of coordination emerge during performance) [5]. Therefore, in such complex systems, team sport athletes must make many decisions as they perceive and interpret the available environmental information relative to the position of the ball, their teammates, and opposing players while executing appropriate actions [6,7]. 

Decision-making is the use of information provided by one’s current situation combined with one’s ability to apply their knowledge about the situation to plan, select, and execute an appropriate goal-directed action or set of actions [8,9]. Decision-making is also considered as the capability of players to choose functional actions from a vast number of possible actions that emerge from the environment to achieve a specific goal [10]. Thus, accurate decision-making has been identified as an important factor for successful performance in team sports [11]. However, it is hypothesized that the quality and accuracy of decisions can be influenced by different covariables, such as age, the relative age effect, or expertise [12,13], as well as acute factors, such as fatigue [14].

Among the abovementioned factors that may influence decision-making, the paramount importance of expertise in the accuracy and type of decisions made in team sports has been demonstrated [15,16]. Comparisons between expert and non-expert players suggest that the superior abilities of experts allow them to make accurate decisions faster than their non-expert counterparts [11,17]. From a cognitive perspective, the higher performance of expert athletes mainly depends on internal mental representations and on the cognitive processes that mediate the interpretation of a stimulus and the selection of an appropriate response [18]. In a recent review with a meta-analysis [16] that aimed to quantify differences among experts and nonexperts (additionally considering concurrent factors, such as the competitive level, age, or skill level), superior cognitive functions were found in experts, as well as the importance of skill to differentiate the cognition–expertise relationship. In this review, however, age seemed to not be significant in differentiating the players [16].

With the purpose of stimulating the quality and accuracy of decisions in youth athletes (aiming to benefit the expertise levels), the proposal of intervention programs for decision-making has been researched [19,20]. In fact, it is expected that the training of perceptual-cognitive expertise may enhance the decisions made by players and improve the quality of practice promoted by coaches [21]. Commonly, training programs dedicated to decision-making may evolve video observation [22], questioning, and pedagogical models [23] and also the application of specifically designed tasks that may benefit decisions and to develop the perceptual-cognitive levels of players [24]. These strategies aim to improve the expertise of players and increase the capacity to develop talents. A recent systematic review on talent identification described a lack of evidence about the combination of fitness/anthropometry variables (the majority of the studies in talents are centered on these types of measures) with parameters related to the technical/tactical aspects of sports and decision-making [3]. The authors mentioned that integrating decision-making skills and tactical behavior into talent identification processes might improve the capacity of coaches to foster the development of their players [3]. Thus, over the past decade, due to the importance of decision-making, the decision-making abilities of youth athletes have received increased attention from the scientific community, and the body of literature investigating the decision-making abilities of youth athletes has grown in many sports [25,26,27,28]. Specifically, researchers have studied several decision-making-related variables, such as decision time [25,26] and response accuracy to a stimulus [27,28]. Other studies have evaluated the effectiveness of several programs in enhancing players’ decision-making abilities [29,30]. 

Along with the growing number of published articles about decision-making, some narrative and systematic reviews have been published [31,32]. One systematic review focused on the effects of expertise on decision-making, revealing that the decision time and response accuracy influenced the magnitude of the difference between novices and experts [31]. Meanwhile, a narrative review presented the models of teaching tactical skills, [33] and another review described the roles of attention, anticipation, and memory in the decision-making process [34]. Despite the importance of the abovementioned reviews, there is an absence of systematization about the type of decision-making studies conducted in youth team-sports players. In team sports, the tactical behavior and the quality of the performance are closely related with the decisions made by players, and for this reason, it is extremely important to understand how decision-making occurs. Moreover, it is also important to qualify the studies conducted in youth and the main topics and evidence that have been produced. Such systematization may help researchers and coaches to improve their understanding about decision-making in youth team-sport players and reorganize practices or strategies. For these reasons, the aim of this systematic review conducted in the topic of youth team-sports players was three-fold: (i) Analyze the variations of the decision-making processes between young players with different levels of ability; (ii) analyze the variations of the decision-making processes between different age groups; and (iii) analyze the effects of decision-making training-based programs on youth players. It was hypothesized that high-level and older players would be more effective in decision-making, i.e., with a greater response accuracy, higher verbalized knowledge, and greater visual searching strategies. Hence, it was hypothesized that decision-making programs improve decision-making in the game, thus improving the player’s performance.

## 2. Materials and Methods 

This systematic review protocol was registered at the International Platform of Registered Systematic Review and Meta-analysis Protocols under number 202040207 and DOI 10.37766/inplasy2020.4.0207.

### 2.1. Search Strategy: Databases and Inclusion Criteria

This systematic review—and the searches associated with it—were performed following the preferred reporting items for systematic reviews and meta-analyses (PRISMA) recommendations [35]. The electronic databases of PubMed, ScienceDirect, Academic Search Complete, SPORTDiscus, and Taylor & Francis Online database were used, as they represent the databases with the largest repository of studies in the field of sports. The search included studies from January 1980 till January 2020 for relevant publications using the following keywords: (youth OR maturation) AND decision-making AND (team sports OR volleyball OR soccer OR futsal OR American football OR football OR basketball OR handball OR rugby OR cricket OR korfball). 

The inclusion criteria were: (i) Decision-making studies in youth players; (ii) decision-making studies in team sports; (iii) only studies comparing different performance levels, players from different age groups, and the effects of decision-making training programs were included; (iv) only full-articles; (iv) only written in English; and (v) the article presented enough information about the sample and experimental approach/procedures (e.g., description of the procedures of data collection, the experimental approach, the instruments, and the measures). The exclusion criteria included: (i) Decision-making only in senior players; (i) studies not related to the specific field of decision-making in team sports; (iii) studies not in sports; (iv) studies not covering the main objectives of comparing different performance levels of youth players, age groups, or analyzing the effects of decision-making-based training programs; and (v) articles with a severe lack of information in methods, not allowing an understanding of the experimental approach and procedures (bad quality). The search was limited to original articles published online until January 2020. Literature reviews, overviews, conference proceedings, and masters and PhD thesis were excluded.

Two of the authors individually screened the citations and abstracts to identify articles that could be included in the review. If an article was deemed potentially useful, the full article was searched for, retrieved, and independently screened by the same two authors to determine whether they met the inclusion criteria. Disagreements between the two authors in terms of the inclusion criteria were resolved by the third author.

### 2.2. Quality of the Studies 

Methodological quality was assessed using the STROBE Statement, which is a 22-item checklist that is considered essential for the accurate reporting of observational studies. This checklist includes a link between the title of the article and its abstract (item 1), introduction (items 2 and 3), methods (items 4 to 12), results (items 13 to 17), and discussion (items 18 to 21) sections, as well as any other information (item 22). Of those, 18 items are common to all three designs, while four of them (items 6, 12, 14, and 15) are design specific, with different versions for all or part of the item. For some items (indicated by asterisks), the information should be given separately for cases and controls in case-control studies or for exposed and unexposed groups in cohort and cross-sectional studies.

Each article was classified based on the sum of the points for all 22 items (one point was counted for an item if the criteria were met), and the result was divided by the maximum possible point total of 22 (e.g., if an article had 11 points, the calculated value was 0.5). The items of all articles were independently classified by each of the observers. Afterwards, an interobserver reliability analysis was conducted. The Kappa index test revealed a value of 0.94 (90% IC: 0.92–0.96), indicating excellent agreement between observers. The results are shown in Table 1.

### 2.3. Low- vs. High-Performance-Level Youth Team-Sports Players

Nine studies particularly compared the decision-making processes between different ability levels of youth team-sports players (Table 2). Considering the methodologies to compare players, two of the studies were more related to visual searching strategies [36,37]. One study [38] was more dedicated to a comparison of verbalized knowledge. Six of them compared the movement and response accuracy of the youth players [6,27,39,40,41,42]. Five of the studies were conducted in association football (i.e., soccer), two in handball, one in Australian football, and one in baseball. Ages represented in these comparisons varied from a minimum of 7 years to a maximum of 17 years. A total of 661 men and 27 women players were analyzed in all the included studies, and 338 were both (not being reported the N for sexes within the studies).

### 2.4. Extraction of Data

For the articles included in this study, all authors discussed how the information should be organized regarding the characteristics of the studies and the results of the assessed measurement properties. Afterwards, two of the authors extracted data regarding the participants’ characteristics (i.e., number, age, and skill level), the study’s objective, its design (i.e., its structure), the measures assessed, and the main results.

## 3. Results

### 3.1. Search, Selection, and Inclusion of Publications

The initial search yielded 6215 titles. The data were exported to Mendeley’s Reference Manager software (Mendeley Desktop, version 1.19.4, Elsevier, London, UK). Replications (199 references) were eliminated manually. The remaining 6016 articles were then screened for relevance based on their title and abstract; 5018 studies were eliminated from the database during this step, mainly because they included studies outside the sports field (mostly economics and psychology). The full texts of the remaining 137 articles were retrieved and examined in detail. From those full texts, 111 were rejected because they did not meet the inclusion criteria. The reasons for exclusion were: (i) Studies not related to team sports and youth players (N = 99), (ii) studies not conducted in team sports (N = 10), (iii) studies not reporting enough information about the experimental approach and procedures (N = 1), and (iv) studies not published in English (N = 1). At the end of the screening procedure, 24 articles were selected for in-depth reading and analysis (Figure 1).

### 3.2. Quality Assessment

The quality of each study is expressed within Table 1, Table 2, and Table 3 included in the results. The minimum and maximal values were 0.68 and 0.86, respectively. Four of them were between 0.6 and 0.7, 14 of the between 0.7 and 0.8, and 7 of them were between 0.8 and 0.9. The mean and standard deviation of all the studies was 0.78 ± 0.06. 

### 3.3. Data Organization

The results are presented in the following three sub-topics: (i) Comparisons of decision-making between low- and high-ability levels in youth team-sports players (nine articles included); (ii) comparisons of decision-making between different age groups in youth team-sports players (seven articles included); and (iii) the effects of decision-making-based training programs in youth team-sports players (10 articles included). Data organization respected the three main objectives defined for the present systematic review. Table 1 presents a summary of the studies included in this systematic review regarding the sports, gender, characteristics of the sample, and quality score from methodological analysis.

### 3.4. Comparisons between Different Age Groups 

A description of the four included studies can be found in Table 3. In the included studies, one was focused on visual searching strategies [43], one on verbalized knowledge [46], and five of them on movement and response accuracy [25,40,44,45,47].

Five studies were conducted in association football (i.e., soccer), one in handball, and one in baseball. Ages varied between a minimum of 7 years old and a maximum of 47 years old (in a specific study [43] that compared different age groups from youth and senior levels). A total of 642 men and 2 women youth players were analyzed in these included studies. 

### 3.5. Effects of Decision-Making-Based Training Programs

Of the 10 studies that analyzed the effects of training programs on decision-making (Table 4), 4 of them used a nonlinear pedagogy-based program to induce changes in the decisions made by participants [29,30,51,52]. Three of the studies used immersive three-dimensional videos to stimulate the decisions made by participants [22,53,54], and two of the studies used video feedback and questioning [48,49]. One study assessed the effects of imagery [56]. Finally, one study compared the effects of implicit and explicit training methods on penalty kicking performance [50]. 

Of the included studies, five were conducted in football (soccer), two were conducted in basketball, one was conducted in futsal, two were conducted in volleyball, and one was conducted in handball. Considering the sexes of the participants, five studies included only boys, three studies included only girls, one study included boys and girls, and two studies did not report the sex of the participants. A total of 94 men, 48 women, and 58 sex-not-defined players were analyzed in the included studies.

The Game Performance Evaluation Tool (GPET) was the most common instrument used to measure decision-making [29,30,51,52]. The period of the training programs varied from 3 [22] to 11 weeks [48,49].

## 4. Discussion

The purpose of this systematic review was three-fold: (i) Analyze the variations of decision-making processes between low- and high-level youth players; (ii) analyze the variations of decision-making processes between different age groups; and (iii) analyze the effects of decision-making training-based programs on youth players. As hypothesized, our results confirmed that high-level players, as well as the older ones, presented a greater accurate response and were more adjustable to complex situations, showing better tactical knowledge and behavior. Regarding decision-making programs, the studies included suggested a beneficial effect when using practical scenarios, confirming our second hypothesis. 

### 4.1. Low vs. High Performance Level of Youth Team-Sports Players

Comparisons between high- and low-performance-level players in youth categories are relatively difficult to conduct (mainly because of the difficulty in characterizing the meaning of being a high- or low-level performance player). In the included studies, comparisons between high- and low-level ability players, more and less talented players, and selected and non-selected players were conducted. Regarding the last one, selected and non-selected football players for football schools were compared in a study conducted in under-11 players [38]. The authors [38] tested game-reading by applying the skill theory coding system to classify the verbalization of players made during the visualization of three selected videos of offensive game plays with a clear starting point and ending point. The skill theory coding system allowed the researchers to identify the complexity level of the player considering the lowest score (0 error) to the abstraction (7), passing by the 1 (single sensorimotor characteristics), 2 (sensorimotor mappings), 3 (sensorimotor systems), 4 (single representations), 5 (representational mappings), and 6 (representational systems) [38]. The results revealed that highly skilled players (selected players) structured the information from the game plays at higher levels of cognitive complexity than non-selected players, suggesting that the instrument may be used to discriminate the level of the players. Moreover, selected players were able to discriminate more game elements than non-selected players, thus suggesting a better capacity to understand the dynamics of the game and establish interactions between its elements [38].

Using a different approach to characterize talent players, a decisions score methodology was employed in two studies [27,41]. Both studies followed similar methodologies [27,41]. Specifically, attacking video clips from the sport (football and Australian football) were given to three expert coaches, who used the clips to rate players’ decisions. After, players watched the videos and provided their scores. The decision-making scores of players were then used to compare talented and non-talented players in Australian football [41] and sub-elite, state elite, and national elite players in football [27]. In the case of Australian football, the decision-making scores were used to identify talented players after watching the clips [41]. In fact, it was possible to observe a discrimination of 92% of talented players and 76% of non-talented players [41]. Similarly, in the study conducted in football (soccer), evidence was found that national elite players had greater percentages of decision-making scores when compared to state elite and sub-elite players [27]. As mentioned by the authors [27], it is expected that differences between players could be exacerbated, considering the exposure to different training scenarios and selection processes that resulted in the preliminary classifications of the groups. In summary, both studies were consistent in highlighting that more talented players [27,41] tend to present meaningfully better decision-making scores than less talented ones. However, further research should be done using practical scenarios while trying to match cognitive and execution decisions.

Considering the ability level of players, two of the included studies [39,40] compared low- and high-ability-level football and baseball players. Both studies included female and male players. In the study conducted in low- and high-level football players (7 to 14 years old) [39], the cognitive and execution dimensions of decision-making in small-sided games in 2 vs. 2 to 7 vs. 7 formats were compared. Regardless of their age group, high-level players revealed better results in the cognitive dimensions of in-game performance [39]. In the particular case of global adaptation to the tactical context of the game, the high-level players were always meaningfully better. Additionally, an increase was found in the gap between low- and high-level players with the progression of age [39]. Finally, in the categories of passing and keeping the ball, the high-level players presented better results than low-level players [39]. 

French et al. [40] aimed to compare differences in the skill executions between baseball players of different ability levels. They revealed that highly skilled players had meaningfully greater throwing distances, batting averages, batting contact, and catching in 7-, 8-, 9-, and 10-year-old players when compared to low-skilled players. However, in the model, the cognitive components of performance contributed minimally to expertise, supporting the idea that these cognitive components are more prevalent in high tactically demanding games than in baseball [40]. In the same study, the results also suggested that experience plays a large role in developing expertise, as moderate correlations between years of practice and skill execution were found only in the older age groups [40]. Despite the small number of studies comparing low- and high-level players [39,40], both of those reported here indicate that the effects of expertise may be higher in older age groups than in younger age groups and that the years of accumulated experience and stimuli may increase the gap between low-level players over the years.

Briefly, it is possible to highlight that more talented players tend to be better at discriminating more game elements than non-talented players as well as being capable of getting greater percentages of decision-making scores when compared to non-talented players. Considering the ability level, it seems that high-level players are more adjustable to new contexts of the game and also have greater cognitive performance.

### 4.2. Comparisons between Age Groups

The comparison of decisions made by players of different ages is one of the most explored topics in decision-making in youth team-sports players. It is expected that some decision-making strategies may be meaningfully different between ages, considering the effects of maturation and years of practice. Considering the visual searching strategies of players from different age groups, Schorer and Baker [43] showed that age played an important role in perceptual-cognitive skills, as better handball goalkeeper performances were observed in senior and expert players than in lower level players. Nevertheless, no difference was observed between senior and expert players, suggesting that the difference between them was only in physical fitness. This effect of age was also noticed in soccer players by Ward et al. [57]. 

Although six studies included both sexes, only two of them really compared decision-making between sexes. In fact, women’s intuitive decisions are often reported to be superior to those of men, due to a superior empathic ability of vicarious emotional responding and nonverbal decoding abilities. However, evidence is still quite limited. In the study of Raab et al. [42], only quite small differences were noticed between sexes, with females presenting a slightly higher preference for intuition than males. On the other hand, in the Panchuck et al. [22] study, any difference was observed in the female group, in an intervention in basketball. Nevertheless, this study presented several methodological flaws (as there were a very small number of females in the control group, only one evaluation was conducted on the post-test, and the amount of training differed between groups), precluding a clear conclusion. Thus, in future studies, differences between sexes should also be considered for analyses.

In a study comparing the verbalized knowledge of different age groups [46], possible relationships between players’ decision self-efficacy (assessed by a Likert-based questionnaire of 10 items) and the options of decisions made were analyzed for a given scenario presented by s video (in which players generated options to solve the scenario). The results of this study [46] revealed that younger players had greater values of decision self-efficacy. Comparisons between age groups also provided evidence that time pressure boosted the decision-making performance in older players [46]. Moreover, considering time pressure, fewer options were made. Interestingly, options selected under time pressure were better than options selected without time pressure [46].

In terms of comparisons of the movement and response accuracy between age groups, a study that compared the cognitive and execution dimensions of decisions during real games played by football players from different age groups [25] revealed that the under-19 group had the highest values for pass actions, while the under-10 group had the lowest. The under-19 group also had the highest percentage of suitable decisions for dribbling, although the under-12 group presented the best decisions for shooting. In the same study [25], a progressive improvement was found from the under-10 to under-12 group but was interrupted in the under-14 group. In terms of executions, similar evidence was found, with a progressive improvement being reversed in the under-14 group. The authors suggest that some possible causes for this are the format of play used in the different age categories, thus speculating that 9 vs. 9 is a better format for the age category than 11 vs. 11 [25]. In the same study [25], the analysis of the relationships between decisions and executions revealed that the correlations increased as players’ age increased, suggesting the importance of experience in making decisions quickly and executing actions properly.

Trying to describe the contexts in which decisions occur in youth football, González-Víllora et al. [47] used the GPET to classify the decision-making processes of players in small-sided games and real formats of play. Comparisons between different age groups revealed that younger players (8 to 10 years old) carried and dribbled the ball more frequently than older players (12 to 14 years old), while older players made more passes. Moreover, older players revealed greater precision in making decisions during games [47]. Additionally, better defensive performances were found among older players, thus suggesting that the exposure of players to specific objectives and task demands should be individualized and adjusted to their age category and experience.

Other actions beyond declarative and execution of technical actions were compared between age groups. In the included studies, Machado et al. [45] and Gonçalves et al. [44] compared the tactical knowledge and behaviors of youth football players from different age groups. Machado et al. [45] identified the tactical skill levels of under-15 and under-17 players using System of Tactical Assessment in Soccer (FUT-SAT) during G+3 vs. 3+GK games. After, players participated in different small-sided games of different difficulty levels, through which the players’ exploratory behaviors were measured using the offensive sequences characterization system. Furthermore, the patterns of play were assessed using lag sequential analysis [45]. The main findings of the study were that older age groups had better tactical skill levels and presented greater exploratory behaviors in the games than younger players. Moreover, less difficult games were better for improving the overall team’s performance and increasing players’ exploratory behaviors [45]. The main implications suggested by the authors [45] are (i) the importance of standardizing practice scenarios according to the tactical skills of players and (ii) individualizing the training process in accordance with their needs and using a system to classify the difficulty of the exercise to easily adjust it to the players.

Tactical skills can be part of the interpretation of decisions made by players. However, tactical skills can also be related to other covariables. A study conducted in under-10 and under-15 soccer players [44] tested the effect of maturational status on peripherical perception and the influence of perception on the efficiency of tactical behavior. The findings revealed that more mature players presented better results in terms of peripheral visual perception. Furthermore, the results also indicated that maturation was moderately to largely correlated with peripheral visual perception measures. The results also showed that better and larger peripheral visual perception was positively related to tactical efficiency. The authors [44] highlighted that maturation and the proficiency of peripherical perception are closely related and help players to identify teammates who are in an advantageous position to receive the ball, assertively predict the next position of the teammates, and better judge the expectations or search for new information without central vision, among other benefits.

Briefly, and despite the different methodologies and approaches used to analyze the effects of different youth age groups on decision-making, it was possible to observe that older players benefit from time pressure during verbalized knowledge assessments and tend to execute more passes and make a greater percentage of suitable decisions during game situations than younger players. Finally, older players tend to present more exploratory behaviors and better tactical knowledge than younger players. 

### 4.3. Effects of Decision-Making-Based Training Programs

Among the 10 studies included in this section, 4 of them used nonlinear pedagogy and training scenarios based on small-sided games to improve the cognitive and execution decisions made by youth futsal and football players [29,30,51,52]. The study carried out by Pizarro et al. [30] applied a training program twice a week for six weeks in under-16 futsal players. The decisions made by players were analyzed using the GPET for cases of passing, dribbling, and shooting [30]. The results of the study showed meaningful improvements in the execution of passes and cognitive passing decisions in situations where the objective was to keep possession of the ball and progress towards the goal. However, no improvements were found in the shooting scenarios [30]. In fact, passes were the main variable of improvement, possibly benefiting from the games in the numerical superiority of attack to maintain ball possession and progress on the field, as suggested by the authors [30].

Additionally, by analyzing the effects of a 14-week nonlinear pedagogy training program (organized into two sessions per week for seven weeks) conducted in low- and average-skill-level under-11 players, it was seen that pass execution and cognitive passing decisions were meaningfully improved in numerical superiority and numerical equality games (in the case of the average-skill-level group) [51]. Interestingly, low-level-skill players improved only from period 1 to period 2 of the intervention, suggesting that low-level players take longer to acquire the conditions necessary to improve the decisions for passes [51]. When comparing a nonlinear pedagogy and a conventional direct instruction program (twice a week over seven weeks), the cognitive and execution decisions for passes were improved in the nonlinear pedagogy group, although no differences were found for dribbling.

In summary, it is possible to conclude that studies are consistent in identifying that nonlinear pedagogy training programs are effective in improving the cognitive and execution decisions for passes. This could be based on the small-sided and conditioned games that involve creating situations of numerical advantages and the circulation of the ball in attacking moments. Interestingly, across the studies that tested the nonlinear pedagogy, the improvements made in passes were not apparent for dribbling. This suggests that training programs do not provide the necessary affordances to increase the judgments needed to make advantageous dribbling decisions. It could be beneficial for future training programs to use descriptive studies that inform players of the reasons for dribbling [58] and the natural scenarios in which such actions occur [59]. In this way, small-sided and conditioned games can be designed so that they augment the perceptions of players for such decisions.

The use of two-dimensional (2-D) and three-dimensional (3-D) videos has been explored to enhance the decision-making of youth players. The assumption is that the use of simulations of sport-related scenarios may improve the perceptions of players and promote a transfer for practical scenarios [60]. In the current systematic review, we identified five studies that used videos/simulated scenarios as part of the training programs used for enhancing decision-making [22,48,49,53,54].

The training programs presented in the studies of Gil et al. [48,49] used a three-step sequence in each session that consisted of: (1) Watching the attack action (video feedback; watching the full point and the selected actions, waiting a moment to favor a stimulated recall, and watching the same action again); (2) performing a self-analysis and reflection (explaining and valuating the selected point and analyzing the decision made); and (3) executing a combined player-expert analysis (questioning, performing a sequential analysis of the causes and reasons throughout questioning). In both studies [48,49], the training intervention lasted 11 weeks and was used to compare the experimental group and the control group.

The studies conducted in female volleyball players [48] and male basketball players [49] revealed within-group changes after an intervention. Specifically, positive improvements were seen in the successful decisions of volleyball players [48], and more successful decisions and executions were observed among basketball players [49]. Both studies confirmed the beneficial effects of a program that combined video feedback and questioning [48,49]. Despite this, the positive effects of the intervention phase were partially lost for the retention, suggesting that decision-making should be consistently worked on throughout the season.

The use of immersive videos to develop the perception of players and increase their exposure to potential real scenarios is one of the great opportunities of 3-D instruments. Comparing the use of 2-D and 3-D videos in a six-week training program in female handball players [54], it was observed that both approaches (2-D and 3-D) improved players’ decision-times (the 3-D approach was slightly more effective). Despite this, no significant benefits were found in terms of the quality of the options selected by players [54]. Possibly, 3-D videos allow players to put themselves in the game situation, justifying the contribution to accelerating the decisions [54]. However, the absence of significant improvements in decision quality suggests that additional strategies should be conducted.

In the same article, the authors [54] conducted a second study in male handball players to compare the effects of a 3-D training program by assessing a group that received instruction in a tactic board and a control group. The results confirmed that meaningful improvements were seen in the 3-D group by decreasing the time needed to make decisions. Interestingly, in both the 3-D and tactic board groups, improvements in the quality of decisions were observed. This finding suggests that the age group and the experience of players contributes to the benefits derived from dedicated decision-making training programs [54].

Testing the transfer from immersive video training programs to performance in small-sided basketball games [22], a study conducted in male and female under-17 players applied a three-week program (10 sessions in total for females and 12 sessions for males) and compared groups that participated in the programs to control groups. The groups were assessed in an immersive test and during small-sided games. The results revealed that females from the experimental and control groups improved their scores in immersive tests and did not improve their small-sided game (practice) performance. Additionally, neither of the male groups significantly improved from the baseline in the immersive tests or during small-sided games [22].

While the study did not provide evidence of the beneficial effects of immersive training, the authors suggested that future interventions should ensure there is enough variety in scenarios or that the stimuli should be specific to the group (e.g., considering the players’ sex, skill level, and background) [22]. A study on older football players [53] compared a passive control group with active groups who performed 3-D multiple object tracking tasks. Significant improvements for passing decisions were observed in the experimental group after five weeks when compared to the control group. However, no meaningful improvements were observed for dribbling or shooting. Finally, one study tested the effects of imagery training on decisions made by youth volleyball players [55]. The study revealed that the experimental group exposed to three sessions a week over the 8-week period showed meaningful improvements in the decisions related to passes. Possibly, the use of mental imagens concerning execution may be a good complementary practice to include in training routines; however, future studies should be made to confirm the hypothesis.

In short, the studies using immersive videos as training programs provided slight benefits for youth players [22,53,54]. The main benefits are associated with a decrease in the time needed to make decisions. However, trivial to small benefits were found in the quality of the decisions and the execution of actions. Perhaps training programs and 3-D videos should be adjusted and individualized based on players’ skill levels and know-how to test possible links to practical contexts. Moreover, future studies should test the sensitivity of different ages and tactical levels of players to the effects of these types of interventions. Eventually, combining immersive videos with practical exercises can enhance decisions and test the effects of these two types of interventions.

### 4.4. Study Limitations and Future Directions

Due to the lack of studies conducted with similar approaches, it was not possible to provide solid evidence in response to the systematic review objectives. This is one of the limitations that should be considered while reading the conclusions. Despite this, the included articles helped to systematize the state-of-the-art regarding the decision-making in youth team-sports players. Future studies should consider using similar study designs as well as similar assessments aiming to improve the value and the generalization of the findings. 

Regarding the training programs, the lack of consistency in the results across the studies reviewed in this article provide an opportunity for future studies to improve the methodological approach of training programs (e.g., to individualize the immersive scenarios, adjust the process to the skill and tactical levels, or adjust them to the players’ age and the context). Moreover, testing a combination of practical drills based on natural scenarios (augmenting the perception of players for specific challenges and tactical problems) with immersive videos is the next logical step for training interventions.

## 5. Conclusions

Comparisons between high- and low-level players revealed that high-level players present greater accuracy in the cognitive and executive answers to the game as well as being more adjustable to more complex situations. Considering the comparisons between age groups, a tendency for older players to execute more accurate decisions in the game and to have better tactical knowledge and behavior was observed. Finally, the effects of decision-making training programs suggest a beneficial effect from employing practical scenarios (mainly based on small-sided and conditioned games), primarily improving passing decisions and execution. However, the benefits of interventions using videos are not clear. As practical implications, this systematic review may suggest that there are clear differences in the decision-making processes between low- and high-ability levels as well as between age groups. For this reason, coaches should consider identifying specific measures that may allow them to quickly identify those changes, aiming to adjust specific interventions based on the player’s needs. Possibly, adjusting training scenarios and task complexities would also be useful considering the different levels of decision-making. Finally, the introduction of specific tasks/programs to develop decision-making should consider a combined use of both video analysis/questioning and drill-based activities. 

## Figures and Tables

**Figure 1 ijerph-17-03803-f001:**
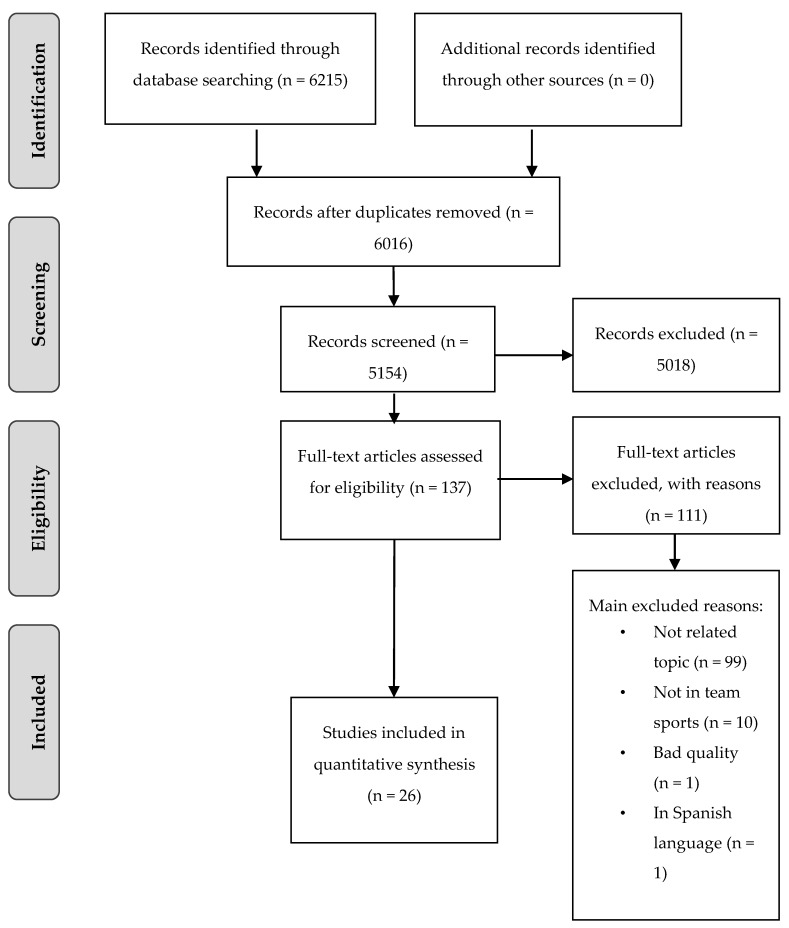
PRISMA flow diagram.

**Table 1 ijerph-17-03803-t001:** Summary of the studies included in the systematic review.

Study	Team Sport	Gender	N and Characteristics of the Sample	QS
Raab et al. [36]	Handball	Female and male	69 high-level players (M: n = 19; F: n = 10), medium-level (M: n = 13; F: n = 9), and low-level (M: n = 8; F: n = 10)	0.77
Vaeyens et al. [37]	Football	Male	n = 87, 13.0–15.8 yo (1) *elite* (n = 21; 14.7 ± 0.5 yo); (2) *sub-elite* (n = 21; 14.6 ± 0.3 yo); (3) regional (n = 23; 14.6 ± 0.6 yo) control group (n = 22; 14.5 ± 0.4 yo)	0.86
Den Hartigh et al. [38]	Football	Male	N = 88; 49 of them invited to practice at one of the five regional football schools, age: 10.91 ± 0.3 yo; and 39 non-invited, age: 10.55 ± 0.4 yo	0.73
Diaz del Campo et al. [39]	Football	Female and male	N = 129; from 7 to 14 yo; 55 high-level and 74 low-level players	0.77
French et al. [40]	Baseball	Female and male	N = 159; 28 of 7 yo; 44 of 8 yo; 45 of 9 yo; and 42 of 10 yo	0.68
Keller et al. [27]	Football	Male	N = 62; age: 17.0 ± 0.6 yo	0.77
Woods et al. [41]	Australian football	ND	N = 50; 25 talent-identified with 17.8 ± 0.5 yo; 25 non-talent-identified with 17.3 ± 0.6 yo	0.68
Bennett et al. [6]	Football	Male	N = 328 (n = 119, age = 12.1 ± 2.6 yo; n = 171, age = 13.2 ± 1.7 yo; n = 38, age = 14.2 ± 1.5 yo).	0.82
Raab et al. [42]	Handball	Female and male	54 (27 male and 27 female) handball players (age = 15.27 ± 1.65 yo) of different ability levels (high-level: *n* = 16; medium-level: *n* = 20; and low-level: *n* = 18).	0.73
Schorer et al. [43]	Handball	Male	(1) sub-youth (n = 8, 14.4 ± 0.5yo); (2) youth (n = 5, 16.8 ± 1.1yo); (3) junior (n = 9, 19.2 ± 1.6yo); (4) adult (n = 8, 27.3 ± 5.8yo), (5) senior (n = 3, 46.7 ± 3.8yo)	0.68
Sevil Serrano et al. [25]	Football	Male	N = 186; ages: between 8 and 19 yo	0.73
Gonçalves et al. [44]	Football	Male	N = 54; 18 players with 9.86 ± 0.2 yo; 18 players with 12.87 ± 0.2 yo; and 18 players with 14.89 ± 0.3 yo	0.73
Machado et al. [45]	Football	Male	N = 48; 24 with a 13.06 ± 1.5 yo; 24 with a 16.89 ± 0.1 yo	0.77
Musculus et al. [46]	Football	Male	N = 97; 48 younger, age: 8.76 ± 1.2 yo; 49 older, age: 12.18 ± 0.9 yo	0.77
González-Víllora et al. [47]	Football	Male	N = 57; 14 players with less than 8 yo; 13 players with less than 10 yo; 14 players with less than 12 yo; and 16 players with less than 14 yo	0.68
Gil-Arias et al. [48]	Volleyball	Female	N = 8; age: 14.75 ± 0.70 yo	0.86
Gil-Arias et al. [49]	Basketball	Male	N = 11; age: 12.7 ± 50.65 yo	0.86
Navarro et al. [50]	Football	Female	N = 20; age: 17.3 ± 2.8 y	0.82
Panchuk et al. [22]	Basketball	Female and male	N = 20; age: 17.0 ± 0.6 yo	0.82
Pizarro et al. [30]	Futsal	Male	N = 8; age: 15.38 ± 0.6 yo	0.77
Praxedes et al. [51]	Football	ND	N = 19; from which 10 average skill-level–age: 10.55 ± 0.5 yo and 9 low skill-level–age: 10.66 ± 0.5 yo	0.82
Praxedes et al. [52]	Football	Male	N = 19; experimental group had 10 players with 10.55 ± 0.5 yo and a control group with 9 players with 11.77 ± 0.7 yo	0.73
Praxedes et al. [29]	Football	ND	N = 19; age: 10.63 ± 0.5 yo	0.77
Romeas et al. [53]	Football	Male	N = 23; age: 21.67 ± 0.5 yo	0.73
Hohmann et al. [54]	Handball	Female and male	N = 20; age: born in 1993 N = 30; age: 14.89 ± 0.8 yo	0.77
Fortes et al. [55]	Volleyball	Male	N = 33; age: 16 ± 0.6 yo	0.86

QS: quality score; ND: not defined; yo: years old.

**Table 2 ijerph-17-03803-t002:** Studies that compared the decision-making processes between low- and high-ability levels in youth team-sports players.

Study	Level of Ability/Performance	Objectives of the Study	Design of the Study	Measures	Main Results
Raab et al. [36]	High-level players, medium-level, and low-level handball players	Describe the link between the use of different information search strategies, the subsequent option-generation process, and the resulting choice characteristics in a realistic sports task.	Three groups with different levels of performance were tested in four waves occurred approximately every 6 months over a period of about 2 years. The video test consisted of 15 clips of about 10s each. Also, a video-based head-mounted infrared eye tracker was used.	Information search, option generation, and choice.	The spatial strategy was employed by ~51% of high-level players, ~41% of medium-level players, and ~55% of low-level players. Based on option-generation, a spatial strategy was employed by ~61% of high-level players, ~36% of medium-level players, and ~59% of low-level players. A significantly better quality of the initial option was found when compared to subsequent options for each of the four waves. Significantly more options were generated in the first wave, and significantly fewer were generated in all subsequent waves.
Vaeyens et al. [37]	Elite, sub-elite, regional, and control group levels in football	Examine differences in decision-making skill and visual search strategies across five categories of small-sided, offensive game simulations in soccer (2 vs. 1, 3 vs. 1, 3 vs. 2, 4 vs. 3, and 5 vs. 3) with participants with different experience and skill level.	Participants stood on two pressure sensitive switches and were required to make the correct tactical decision quickly and accurately when the ball was played in the direction of the player wearing the yellow vest. Players were required to verbalize their intended response immediately after each trial.	Decision time, response accuracy, search rate, fixation location, and fixation order.	No significant differences in choice reaction times were observed across groups. The three groups of players employed shorter decision-times than the nonplayer participants across all viewing conditions. All participants were less accurate in the 4 vs. 3 condition and more accurate in the 2 vs. 1 condition. Significant differences were observed between the 2 vs. 1 and 3 vs. 1 conditions and the 3 vs. 2, 4 vs. 3, and 5 vs. 3 conditions. Skilled players spent more time fixating on the player with ball possession and less time on the player wearing the yellow shirt.
Den Hartigh et al. [38]	Invited and non-invited players to be part at a football academy	Compare players with greater and lower ability levels in terms of game-reading	Players watched football game plays and verbalized simultaneously the actions taking place in the field.	The Skill Theory coding system was used to code the verbalizations made by the players. The system presents 8 complexity levels (0–error; to 7–abstraction).	Selected players (invited by football schools) had meaningfully high scores on the skill theory complexity scale. Selected players displayed a strong capacity to structure information from the game plays, indicating high levels of cognitive complexity).
Diaz del Campo et al. [39]	Low and high football players	Analyze differences in decision-making (cognitive and execution) between high- and low-level players.	Different small-sided games were applied in accordance to age group (2 vs. 2 to 7 vs. 7). Decision-making during attacking and defensive processes were analyzed comparing low- vs. high-level players.	The Game Performance Evaluation Tool (GPET) was used to determine the decision-making of players (cognitive and execution). Decisions were categorized relative to technical/tactical skills and relative to tactical context adaptation.	High-level players had better results in the cognitive aspects of game performance (independently of their age group). High-level players made better decisions related to passing and keeping the ball than younger players. Results suggest the importance of adapting to the tactical contexts of the game in the development of expertise.
French et al. [40]	Different cognitive and skill execution baseball levels	Analyze differences in cognitive and skill execution components of the game performance in different levels of ability.	A minimum of 5 regular season games were recorded and analyzed for each team.	The following categories were coded by the observers: (i) setting information; (ii) position played; (iii) type of movement; (iv) position decisions; (v) type of control; (vi) location of play; (vii) accuracy of decision; (viii) skill execution (including infield throwing, outfield throwing, tagging a base, tagging a runner), and (ix) forceful execution of throws.	Differences in skill execution between ability levels were found. However, cognitive components were not meaningfully different between ability levels. Throwing force, batting average, batting contact, and catching meaningfully distinguished ability levels. Cognitive components minimally distinguished the ability levels.
Keller et al. [27]	Sub-elite, state elite, and national elite football players	Analyze if a video-based decision-making task could classify football players into different ability levels. Players were organized as sub-elite, state elite and national elite.	Players watched clips in which was necessary to identify the most appropriate option to pass or shoot.	Decision-making score was measured by each player.	The video-based decision-making tests were able to discriminate different levels of performance. A significant increase in decision-making performance with increasing levels of ability level across the three groups was observed.
Woods et al. [41]	Talent-identified and non-talent-identified Australian football players	Analyze if contextual decision-making skill can be a discriminative of talent-identified junior Australian football players	Players were asked to watch a clip of an attacking action and choose the preferred passing option.	Decision-making score was measured by each player.	Talent-identified participants were more accurate than other players in terms of the decisions they made after watching attacking clips.
Bennett et al. [6]	Different football players levels	Evaluate the use of mobile technology as an alternate method of delivering video-based decision-making assessments for talent identification.	Players completed a video-based decision-making assessment on an iPad, with response accuracy and response time recorded for various attacking situations (2 vs. 1, 3 vs. 1, 3 vs. 2, 4 vs. 3, and 5 vs. 3).	(i) response accuracy, measured on a multiple point scale evaluated by two nationally and one internationally accredited coaches; (ii) response time, recorded as the duration between the occlusion of a video and the player selecting a response on the iPad.	Older players were faster at responding in each situation. However, response accuracy was similar in all developmental stages. Therefore, there is limited conclusive evidence supporting the effectiveness of these assessments for talent identification.
Raab et al. [42]	High-, medium- and low-level handball players	Investigate whether a preference for intuition over deliberation results in faster and better lab-based choices in team handball attack situations.	Athletes were asked to name, as quickly and as accurately as possible the first option for the player in ball possession that came to mind after the frozen frame of video clips from a video test.	It was recorded the verbal responses (dependent variables of decision time), option generation time, quality of first option, final option, and number of options. The PID scale was used.	High-level players showed better performance than medium-level and lower-level. Girls were more intuitive than boys. Athletes classified as having a preference for intuitive decisions made their first choice faster, had a better first option, and had better best options than athletes classified as deliberative decision-makers.

**Table 3 ijerph-17-03803-t003:** Studies that compared the decision-making processes between different age groups in youth team-sports players.

Study	Age-Group	Objectives	Design	Measures	Main Results
Schorer et al. [43]	Sub-youth, youth, junior, adult, senior	Determine whether the perceptual-motor abilities of highly skilled performers in dynamic, time-constrained sports exhibited the same pattern of age-related decline seen in other areas.	Three different tests were conducted with each participant: (i) performed an eye-tracking handball video task; (ii) a temporal occlusion handball video task; and (iii) an eight- choice reaction time task.	Movement initiation time of the goalkeepers, reaction quality, and movement time. Number of fixations as well as relative and absolute fixation durations were counted as additional dependent measures for both eye-tracking tasks. EYE	No significant differences of reaction quality in eye-tracking or for choice reaction time tasks were observed, but differences in temporal occlusion were noticed. Older groups had more and longer fixations than younger groups. Seniors had significantly lower scores than sub-youths, juniors, and adults for response selection measured by eye-tracking.
Sevil Serrano et al. [25]	Age groups from 8 to 19 years old	Analyze the decision-making and execution of football players from different age-groups. Moreover, it was also aimed to test the relationships between decision-making and execution of game actions.	30 games were analyzed.	The Game Performance Evaluation Tool (GPET) was used to determine the decision-making of players (cognitive and execution).	Comparisons of decision-making between age groups did not reveal meaningful differences above the under-14 category. Comparisons between decisions made and executions revealed that independent of their age, players were less successful in the execution than in the selection of decisions. Relationships between correct decisions and successful executions were progressively better for older players.
Gonçalves et al. [44]	Groups: 9.86 ± 0.2 yo; 12.87 ± 0.2 yo; and 14.89 ± 0.3 yo	Examine the relationships between maturation and peripheral perception and analyze the influence of peripheral perception on the efficiency of tactical behavior.	In the experiment 1 the peripherical perception was measured using the Vienna Test System. In the experiment 2 players participated in a GK+3 vs. 3+GK format. The efficiency of tactical behavior was measured.	Peripherical perception had the following measures: (i) visual field; (ii) tracking deviation; (iii) reaction time; and (iv) amount of omitted reactions. The System of Tactical Assessment in Soccer (FUT-SAT) was used to determine the success rate of offensive and defensive tactical principles.	The maturational process had a beneficial effect on peripheral visual perception. Maturation had moderate-to-large correlations with the majority of peripheral measures. The size of the visual field influenced the efficiency of offensive behaviors, and the number of omitted reactions affected players’ defensive and overall efficiency.
Machado et al. [45]	Groups: 13.06 ± 1.5 yo; and 16.89 ± 0.1 yo	Analyze the influence of tactical skills level and age category on players exploratory behavior in tasks with different difficulty levels.	Players were assessed for their tactical skill level in a GK + 3 vs. 3 + GK format. The following tasks were then employed: (i) high difficulty small-sided and conditioned game (GK + 4 vs. 4 + GK); and (ii) low difficulty small-sided and conditioned game (GR + 3 vs. 3 + GK + 3 floaters)	The System of Tactical Assessment in Soccer (FUT-SAT) was used to determine the tactical skill level of players. The offensive sequences characterization system and lag sequential analysis technique were used to determine the team’s tactical performance and player’s exploratory behavior.	Teams with players with higher tactical skill levels presented better performance and more exploratory behavior than other teams. Moreover, older players performed better than younger players and exhibited more exploratory behaviors. Tasks with lower difficulty levels promoted better team performance and more exploratory behaviors.
Musculus et al. [46]	Groups: 8.76 ± 1.2 yo; and 12.18 ± 0.9 yo	Analyze the link between developing players’ decision self-efficacy and their decision-making processes comprising option generation and selection.	Players were inquired about their decision self-efficacy and were tested for their capacity to generate options during the visualization of football video-scenes. Manipulation of time pressure was used.	Decision self-efficacy was assessed using a 10-item questionnaire. Decision-making test consisted in to use video-scenes of live football with temporal occlusions. During the stops of the video, the players generated their options.	Younger players revealed a higher decision self-efficacy than older players. Results did not provide strong evidence for the relationship between self-efficacy and decision-making (for both groups). Motor confidence was related to self-efficacy. Older players generated faster and better first decisions than younger players.
González-Víllora et al. [47]	groups: less than 8 yo; less than 10 yo; less than 12 yo; and less than 14 yo	Describe the nature of decisions during the game, compare the solutions provided by players in different scenarios and analyze the relationships between decision-making and skills execution.	The following small-sided games were implemented, recorded and analyzed: (i) 2 vs. 2; (ii) 3 vs. 3; (iii) 5 vs. 5; and (iv) 7 vs. 7+GK.	The Game Performance Evaluation Tool (GPET) was used to determine the decision-making of players (cognitive and execution).	Comparisons between extreme age groups (10–11 and 16–17 yo) revealed that the youngest group had higher precision; however, no differences between closer age groups were found. In the under-8 and under-10 age groups, greater frequencies of carrying the ball and dribbling were observed. The older age groups (12 and 14 yo) made more passes than younger age groups.
French et al. [40]	Groups: 7 yo; 8 yo; 9 yo; and 10 yo	Analyze differences in cognitive and skill execution components of the game performance in different levels of ability.	A minimum of 5 regular season games were recorded and analyzed for each team.	The following categories were coded by the observers: (i) setting information; (ii) position played; (iii) type of movement; (iv) position decisions; (v) type of control; (vi) location of play; (vii) accuracy of decision; (viii) skill execution (including infield throwing, outfield throwing, tagging a base, tagging a runner) and (ix) forceful execution of throws.	Comparisons between ages revealed that throwing force, batting average, and years of experience were discriminants.

yo: years old.

**Table 4 ijerph-17-03803-t004:** Studies that analyzed the effects of training programs on decision-making in youth team-sports players.

	Groups Included	Objectives	Design	Measures (Pre-Post)	Main Results
Gil-Arias et al. [48]	Control and intervention	Compare the effects of video-feedback and questioning training program on decision-making	Two groups were implemented by 11 weeks (N = 4, experimental group; N = 4, control). The study had three phases: (i) pre-test; (ii) intervention; (iii) retention. Intervention occurred in a 6 vs. 6 format, viewing the attack action, self-analyzing and making reflections about the attack and combining analysis of a player-expert.	Seven categories were analyzed during attack actions, using the Game Performance Assessment Instrument (GPAI): (i) base; (ii) adjust; (iii) decision-making; (iv) skill execution; (v) support; (vi) cover; and (vii) guard/mark	Between-group comparisons revealed significant changes during the intervention (*p* = 0.015; ES = 0.652) but not during the pre-test or retention stages. Within-group changes occurred in the experimental group when comparing the pre-test and intervention stages (*p* = 0.041) and the intervention and retention stages (*p* = 0.003).
Gil-Arias et al. [49]	Control and intervention	Compare the effects of video-feedback and questioning training program on decision-making	Two groups were implemented by 11 weeks (N = 5, experimental group; N = 6, control). The study had three phases: (i) pre-test (5 matches); (ii) intervention (11 matches); (iii) retention (5 matches). Players in the experimental group spent 45 min/session on the decision training, viewing the attack action, self-analyzing and making reflections about the attack and combining analysis of a player-expert.	The French and Thomas observation instrument were used to classify the decision-making as succeeded or non-succeeded. Technical actions were also classified as succeeded or non-succeeded. A questionnaire was used to assess the procedural knowledge.	The experimental group showed meaningful improvements in the mean percentage of successful decisions and executions when compared to the control group during the retention phase. However, no meaningful changes were found during the retention phase. Furthermore, no meaningful changes were found between groups in the pre-test or intervention stages in terms of declarative knowledge.
Navarro et al. [50]	Two intervention groups	Compare the effects of implicit and explicit training methods on penalty kicking performance	Participants were assigned to two groups: (i) low-saliency group (changes in task-difficulty were gradual) and (ii) high-saliency group (changes in task-difficulty were large. The practice consisted in three sessions.	Sixty kicks (30 in low-pressure and 30 in high-pressure) on the projected target circle were executed by players. Percentage of goalmouth hits, percentage of target hits, accuracy and percentage of kicks to correct side were analyzed.	Taking less time to make a decision decreased penalty kick performance. Both groups displayed excellent performance when more than 850 ms were given to decide and execute the kick. Implicit and explicit training methods resulted in similar levels of decision-making. However, implicit training increased kicking accuracy.
Panchuk et al. [22]	Control and intervention	Analyze the effects of immersive video on decision-making performance and transfer to passing performance in small-sided games.	Two groups were assigned: (i) training group (viewed 15 randomly selected immersive videos prior to the regular training session); and (ii) control group (only participated in training sessions). Training intervention lasted 3 weeks. Females had 10 intervention sessions and males 12.	Scores at immersive test (coaches classified the decisions made during the videos and this standard was compared with scores made by players) Scores at small-sided games depended from successful pass, hockey assist, assist, open shot, contested shot, deflected pass/bobble, passing turnover and dribble turnover.	In females, the control and intervention groups significantly improved their scores on an immersive test. In males, no group showed significant improvements despite a large magnitude of change in the intervention group. Considering performance in small-sided games, females did not significantly improve in either group (control and intervention) even though a large magnitude of changes were observed in the control group. Moreover, neither group of males significantly improved even though the magnitudes of changes in the intervention group were medium to large.
Pizarro et al. [30]	Intervention	Analyze the effect of a nonlinear pedagogy training program on the decision-making and execution of passing, dribbling and shooting.	Training intervention (nonlinear pedagogy) occurred in 12 training sessions distributed by 6 weeks. No control group was used.	Decision-making and execution were measured as the percentage of successful decisions/executions over the total number of decisions/executions made during matches. The Game Performance Evaluation Tool (GPET) was used to classify the appropriate decision/executions in the following actions: (i) pass; (ii) dribbling; and (iii) shooting.	Decision-making related to passes and the execution of passes significantly improved in the tactical principles of maintaining ball possession and progression towards the goal. No significant improvements were found in the tactical principle of shooting with the lowest level of opposition. Only decision-making related to dribbling significantly improved in the tactical principles of progression towards the goal and shooting with the lowest level of opposition. No meaningful changes in decision-making or executions were found for shooting.
Praxedes et al. [51]	Intervention	Analyze the effect of two training programs on decision-making and technical execution.	Intervention 1: training program of 14 sessions (7 weeks) consisting in modified games with numerical superiority in attack. Intervention 2: training program of 14 sessions (7 weeks) consisting in modified games with numerical equality. All the players participated in both interventions. Intervention 1 occurred firstly and then intervention 2. A pre-intervention 1 and 2 occurred to determine the baseline levels before interventions.	Performance Evaluation Tool (GPET) was used to classify the appropriate decision/executions in matches. Only the passes were analyzed.	The average skill-level group improved their decision-making and execution after intervention 1 (in comparison to baseline–pre-intervention 1). The low skill-level group improved their execution from pre-intervention 1 to intervention 2. The low skill-level group took longer to improve their execution and decision-making. Despite that, numerical superiority did not improve any group.
Praxedes et al. [52]	Control and intervention	Analyze the effect of a nonlinear-based training program in the decision-making and technical execution.	Two groups were compared: (i) experimental group (intervention with a non-linear training program) and (ii) control group (direct instruction). Experimental group made 4 motor tasks per session, each lasting 15 min. Intervention occurred during 14 sessions over 7 weeks	Performance Evaluation Tool (GPET) was used to classify the appropriate decision/executions in matches. Passes and dribbling were analyzed.	The intervention group showed significant improvements in decision-making and the execution of passes when compared to the control group. However, no significant changes were found between groups for decision-making and the execution of dribbling actions. Moreover, the groups did not significantly improve the decision-making and execution of actions related to dribbling.
Praxedes et al. [29]	Intervention	Analyze the effect of a nonlinear-based intervention program on tactical behaviors and decision-making.	All the players started with direct instruction (six sessions for three weeks), moving to nonlinear pedagogy intervention (seven weeks) and finally a retention period of three weeks. No control group was used.	Performance Evaluation Tool (GPET) was used to classify the appropriate decision/executions in matches. Only passes were assessed.	Significant improvements in decision-making and performance behaviors were observed after the intermediate and final points of the acquisition phase. Additionally, significantly higher decision-making and execution scores were reported during retention when compared to baseline (pre-intervention).
Romeas et al. [53]	Control and intervention	Analyze the effect of a three-dimensional multiple object tracking task in decision-making accuracy on the field	The experimental group were actively trained ten times (twice a week for five consecutive weeks) participating in three CORE sessions of 3d-MOT Active control: participants focused on three-dimensional soccer videos (twice a week for five consecutive weeks) Passive control: no instruction or training.	Passing, dribbling and shooting accuracy were assessed. Decision accuracy and subjective decision-making accuracy were analyzed.	Decision-making accuracy in passing was significantly improved from the baseline to the post-intervention period in the 3-D-MOT group when compared to the control group. No significant changes between groups were found for dribbling or shooting.
Hohmann et al. [54]	Control and intervention	Analyze the effectiveness of video-based decision training	The study 1: two experimental groups (2-D and 3-D video group) were employed. Interventions occurred during 6 weeks and a retention test was performed 4 weeks after the end of intervention. The study 2: compare the performance of a 3-D video group with a tactic board group and a control group.	Percentage of correct first and best options, mean decision-time for first option and best option were measured in study 1.	The results of study 1 revealed that 3-D video simulation was slightly more effective than 2-D video simulation (regarding the decision-time). However, neither group improved in terms of the quality of choices made. In study 2, the 3-D group meaningfully decreased the amount of time it took to make a decision (turning faster) in comparison to the control group and tactic board group. The 3-D and tactic board groups revealed slight improvements in the quality of the best option from the post-test to retention.
Fortes et al. [55]	Control and intervention	Analyze the effectiveness of an imagery training program	Effects of 8-week period with three sessions/week were analyzed. Experimental group: imagery intervention during 10-min Control group watched videos of advertisements	The GPAI instrument was used to qualify the decisions.	Moderate positive effect of imagery training on the passing decision-making performance.

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
