# Peer review of "Decision-Making in Youth Team-Sports Players: A Systematic Review"

_ijerph, 2020, doi:10.3390/ijerph17113803_

Round 1

Reviewer 1 Report

ABSTRACT

19-20. Databases could be not correct. (editorial group?).

35. Team sports as a key word?, then title should be related to team sports.

INTRODUCTION

It needs to be longer, deepen the type of decision-making requirements of team sports (time pressure, lack of information, rival action, contextual and situational factors, ...) and the specific characteristics of young players in terms of maturation, psychological development, experience, anticipation, automatisms,...

Decision-making parameters could be inside psychological characteristics?

METHODOLOGY

2.3. Extraction of data:

use two independent reviewers out of the authors?

How did you decide the categorization of the topics? Experts?

RESULTS

PRISMA: I can t understand clearly in the flow chart the reasons for what 5018 records were excluded and the after 136 selected, the reasons for the next exclusion. I guess papers "not in sports" should be removed earlier in the flow chart.

163. eliminate "time"

Tables of results: first column: write the title, author and year; not only the reference

Some of the papers in the results could be in several tables (different topics) at the same time.. 

Ex: Study nº 4 could be in topic decision-time at it is and also in response accuracy.

Ex: nº 32 could be in different topics also.

Others..; please check it.

190: Table 4: There is no other notes under all tables

Numbers of the references in the explanation of the tables are not the same than in the tables. Ex: Table nº4 (34, 16, 17) Text: line 183: (16,17,31)

Then in the next, which also is again table 4.. incorrect

Text (32-34) - Table (35-37)

There is no table 3, and there are two tables 4

Difficult to follow the results. Unclear Please check the references in the text and in the tables, the number of the tables and organize better the studies in the topics or introduce one study in more than one topic.

Lines 174-176; 178,  misquoted references

DISCUSSION

Mostly correct and deeply argued. Just the sections should be reorganized in relation to the results.

The comparison between novice and expert, younger and older are always present in the discussion but not in the results, where most studies have participants of approximate age.

May be this comparison should be included as a aim of the study..

Lines 250, 281, 304, 307,  311:  misquoted references

Lines 353-354. Results of both studies are opposite..

CONCLUSIONS

In some of the topics, its impossible to establish conclusions due to the limitation of the sample and the heterogeneity of the measures and main results.

AUTHOR CONTRIBUTIONS are not filled

REFERENCES

702, 728. Misquoted references.

Author Response

ABSTRACT

19-20. Databases could be not correct. (editorial group?).

  1. Team sports as a key word?, then title should be related to team sports.

AUTHORS: DEAR REVIEWER, THANK YOU. WE HAVE CHANGED THE TITLE.

INTRODUCTION

It needs to be longer, deepen the type of decision-making requirements of team sports (time pressure, lack of information, rival action, contextual and situational factors, ...) and the specific characteristics of young players in terms of maturation, psychological development, experience, anticipation, automatisms,...

Decision-making parameters could be inside psychological characteristics?

AUTHORS: DEAR REVIEWER, THANK YOU. WE HAVE DRASTICALLY CHANGED THE INTRODUCTION AIMING TO IMPROVE THE RATIONAL AND THE BACKGROUND’S QUALITY.

METHODOLOGY

2.3. Extraction of data:

use two independent reviewers out of the authors?

AUTHORS: DEAR REVIEWER, THANK YOU. WE HAVE CHANGED TO “AUTHORS”.

How did you decide the categorization of the topics? Experts?

AUTHORS: DEAR REVIEWER, BASED ON THE MAJOR MODIFICATIONS, THE NEW TOPICS ARE IN LINE WITH THE THREE OBJECTIVES OF THE STUDY.

RESULTS

PRISMA: I can t understand clearly in the flow chart the reasons for what 5018 records were excluded and the after 136 selected, the reasons for the next exclusion. I guess papers "not in sports" should be removed earlier in the flow chart.

AUTHORS: DEAR REVIEWER, THANK YOU. ACTUALLY, WE HAVE UPDATED TO “NOT IN TEAM SPORTS”.

  1. eliminate "time"

AUTHORS: DEAR REVIEWER, THANK YOU. WE HAVE REMOVED.

Tables of results: first column: write the title, author and year; not only the reference

Some of the papers in the results could be in several tables (different topics) at the same time.. 

AUTHORS: DEAR REVIEWER, THANK YOU. WE HAVE ADDED THE NAMES. THE NUMBER OF REFERENCE IT CAN BE USED TO CHECK THE YEAR. THIS IS NOT TO CHANGE THE REFERENCING STYLE OF THE JOURNAL.

Ex: Study nº 4 could be in topic decision-time at it is and also in response accuracy.

Ex: nº 32 could be in different topics also.

Others..; please check it.

AUTHORS: DEAR REVIEWER, THANK YOU. WE HAVE DRSTICALLY CHANGED THIS SECTION.

190: Table 4: There is no other notes under all tables

AUTHORS: DEAR REVIEWER, THANK YOU. WE HAVE ADDED THE NOTES IN ALL THE TABLES.

Numbers of the references in the explanation of the tables are not the same than in the tables. Ex: Table nº4 (34, 16, 17) Text: line 183: (16,17,31)

AUTHORS: DEAR REVIEWER, THANK YOU. WE HAVE UPDATED THE AUTOMATIC REFERENCING SYSTEM.

Then in the next, which also is again table 4. incorrect

Text (32-34) - Table (35-37)

AUTHORS: DEAR REVIEWER, THANK YOU. WE HAVE UPDATED THE AUTOMATIC REFERENCING SYSTEM.

There is no table 3, and there are two tables 4

AUTHORS: DEAR REVIEWER, THANK YOU. WE HAVE CHANGED THE TYPO.

Difficult to follow the results. Unclear Please check the references in the text and in the tables, the number of the tables and organize better the studies in the topics or introduce one study in more than one topic.

AUTHORS: DEAR REVIEWER, THANK YOU. WE HAVE UPDATED THE REFERENCES, ADDED THE NAME OF THE AUTHORS AND BETTER DESCRIBED THE CRITERIA TO ADD AN ARTICLE TO A TOPIC. THANK YOU FOR YOUR FEEDBACKS THAT HELPED US TO RE-ORGANIZE THE SECTION.

Lines 174-176; 178, misquoted references

AUTHORS: DEAR REVIEWER, THANK YOU. WE HAVE UPDATED THE REFERENCES.

DISCUSSION

Mostly correct and deeply argued. Just the sections should be reorganized in relation to the results.

The comparison between novice and expert, younger and older are always present in the discussion but not in the results, where most studies have participants of approximate age. May be this comparison should be included as a aim of the study..

AUTHORS: DEAR REVIEWER, THANK YOU. FOLLOWING YOUR SUGGESTION, WE HAVE CREATED THREE MORE OBJECTIVE GOALS FOR THIS SYSTEMATIC REVIEW AND THE DISCUSSION WAS ORGANIZED BASED ON THAT.

Lines 250, 281, 304, 307,  311:  misquoted references

AUTHORS: DEAR REVIEWER, THANK YOU. WE HAVE UPDATED.

Lines 353-354. Results of both studies are opposite..

AUTHORS: DEAR REVIEWER, THANK YOU. WE HAVE CHANGED THE PARAGRAPH.

CONCLUSIONS

In some of the topics, its impossible to establish conclusions due to the limitation of the sample and the heterogeneity of the measures and main results.

AUTHORS: DEAR REVIEWER, THANK YOU. WE HAVE CHANGED THE CONCLUSIONS AND CREATED A SECTION OF STUDY LIMITATIONS IN THE DISCUSSION.

AUTHOR CONTRIBUTIONS are not filled

AUTHORS: DEAR REVIEWER, THANK YOU. WE HAVE ADDED.

REFERENCES

702, 728. Misquoted references.

AUTHORS: DEAR REVIEWER, THANK YOU. WE HAVE UPDATED THE REFERENCES.

Reviewer 2 Report

ABSTRACT

The abstract should really focus on why we need this study, a broad summary of the method and findings, and what it all means. What we have here is a vague and yet dense description of the main topics in one single study. (a) why was this study needed; (b) what exactly is known already; and (c) what valuable knowledge would be added by doing a systematical review (d) what specific research questions were necessitated in addressing what is 'missing' and (e) what specific hypotheses were formed based on specific theoretical and empirical content? Then summarize the method and findings and give us the take-home message. Remember, you are presenting a systematic review and not a research paper. Thus, “beneficial effect of training programs…” is a highly questionable statement. Consider using practical implications that are obtained thanks to your work.

INTRODUCTION

Page 1, line 40-41: Big claim. citation needed.

Page 2, line 82: The authors say that “to the best of our knowledge, no literature review…” This is such a specific claim, so as to not hold much meaning. It is your own judgment and not expected in scientific writing. There are a lot of unknown questions, however, what is the main focus and why should we examine it? What have the research and sports community to gain with your study?

Most of the introduction section is just describing the basic tenets principals of decision making, not developing a “need” for this study or deriving the hypotheses of the study. This is effectively not doing its job. As a matter of fact, no hypotheses are presented nor do we know what is expected to observe based on existing literature: Are there inconsistencies in previous research?

It seems that little real work was done by the authors to explain why we really need this study and what it would actually add in terms of meaningful value/advancement. We need the introduction to be much more developed.

METHODS

Not being pre-registered with PROSPERO or similar is problematic for such a paper.

The inclusion criteria are vague. The inclusion of studies using young and/or maturation (what does this mean?) will bias results since they present different groups of athletes. The same principle applies to professional and amateur athletes. If the authors do not agree, they need to provide a strong justification for why they included all types of athletes in their systematic review. I suggest citations for their inclusion criteria.

There is clearly 'almost nothing going on' due to the lack of a priori hypotheses.

RESULTS

The authors are recommended to report exclusion criteria and motives for excluding the records in the screening step.

The authors split their remained 35 papers into six subsections. However, the rationale is not provided in the introduction section and, most importantly, this decision limits interpretation. How can we draft conclusions based on decision-time based on n = 3? I continue to not fully understand the hypotheses that are (or not) presented. If they want to maintain their results, information is required.

The authors report “the studies varied between 0.68 and 0.86…” however they do not report cutoffs for each category of study quality. Please insert in the method section

Information in the Tables is extensive and confusing.  The authors are recommended to review their data display so that readers are able to understand what is going on. Additionally, the text should help readers on understanding what is going on in the respective tables.

Still in Table 1, what does “experienced” rugby players mean? Isn’t the study objective youth sport as stated in the title?

At this phase, I truly am having some problems understanding the authors' objective.

DISCUSSION

I will not comment much on the Discussion as I believe it will change considerably if the authors follow my advice on previous sections. Overall, I advise beginning this section by providing the study’s answer to the specific research question (from the introduction section). What did the authors intend to examine?

Then, review findings in relation to previous research and theories – are they consistent, inconsistent? If so/not… then why?

Review implications of findings for (a) existing understanding of the topic of decision-making; (b) future research, research methods; and (c) applied practice in the sport context

Please consider limitations of the study (currently no limitations are presented) and assess the impact/meaning of these limitations when trying to draw inferences/implications.

CONCLUSION

This section will arguably receive a lot of revisions after considering the reviewers' suggestions. Authors are advised to conclude by re-summarizing the findings, and their likely meaning/impact in light of those limitations and offer a clean ‘take-home-message’ that the study findings defensibly justify.

Other comments:

With such big data, a meta-analytical approach is advised. It would increase the quality of the authors’ work and give more robust information on the proposed subject. I understand that this might seem to be a ‘hurdle’ for the authors to get done. However, there are possible issues facing current interpretations.

English writing in the current manuscript is not for publication and requires major improvements. There are frequent grammatical and typographical errors present throughout the paper, which mar the reader’s understanding of the research. Overall, the paper would benefit from a review from a native English speaker or proofread by editing service.

Author Response

ABSTRACT

The abstract should really focus on why we need this study, a broad summary of the method and findings, and what it all means. What we have here is a vague and yet dense description of the main topics in one single study. (a) why was this study needed; (b) what exactly is known already; and (c) what valuable knowledge would be added by doing a systematical review (d) what specific research questions were necessitated in addressing what is 'missing' and (e) what specific hypotheses were formed based on specific theoretical and empirical content? Then summarize the method and findings and give us the take-home message. Remember, you are presenting a systematic review and not a research paper. Thus, “beneficial effect of training programs…” is a highly questionable statement. Consider using practical implications that are obtained thanks to your work.

AUTHORS: DEAR REVIEWER, THANK YOU. CONSIDERING THE VERY IMPORTANT SUGGESTIONS MADE BY YOU AND REVIEWER 1, WE HAVE MADE DRASTIC CHANGES IN THE MANUSCRIPT. THE ABSTRACT WAS ALSO RE-WRITTEN.

INTRODUCTION

Page 1, line 40-41: Big claim. citation needed.

AUTHORS: DEAR REVIEWER, THANK YOU. WE HAVE ADDED.

Page 2, line 82: The authors say that “to the best of our knowledge, no literature review…” This is such a specific claim, so as to not hold much meaning. It is your own judgment and not expected in scientific writing. There are a lot of unknown questions, however, what is the main focus and why should we examine it? What have the research and sports community to gain with your study?

 Most of the introduction section is just describing the basic tenets principals of decision making, not developing a “need” for this study or deriving the hypotheses of the study. This is effectively not doing its job. As a matter of fact, no hypotheses are presented nor do we know what is expected to observe based on existing literature: Are there inconsistencies in previous research?

 It seems that little real work was done by the authors to explain why we really need this study and what it would actually add in terms of meaningful value/advancement. We need the introduction to be much more developed.

AUTHORS: DEAR REVIEWER, THANK YOU. WE HAVE DRASTICALLY CHANGED THE INTRODUCTION AIMING TO IMPROVE THE RATIONAL AND THE BACKGROUND’S QUALITY.

METHODS

Not being pre-registered with PROSPERO or similar is problematic for such a paper.

AUTHORS: DEAR REVIEWER, THANK YOU. WE ENGAGED IN THE REGISTRATION IN EARLY JANUARY. UNFORTUNATELY, SO FAR, WE HAVE NOT BEEN DELIVERED AN ANSWER. IN THE WEBSITE, THEY HAVE EVEN UPLOADED A WARNING STATING THAT THEY ARE VERY DELAYED DUE TO ENORMOUS AMOUNTS OF INCOMING REQUESTS. FURTHERMORE, THEY INFORM THAT ALL UK PROJECTS HAVE PRIORITY. THEREFORE, IT IS CURRENTLY NOT FEASIBLE TO REGISTER THESE WORKS IN PROSPERO, UNLESS WE WISH TO WAIT FOR ONE YEAR OR MORE IN THE WAITING LINE.

The inclusion criteria are vague. The inclusion of studies using young and/or maturation (what does this mean?) will bias results since they present different groups of athletes. The same principle applies to professional and amateur athletes. If the authors do not agree, they need to provide a strong justification for why they included all types of athletes in their systematic review. I suggest citations for their inclusion criteria.

AUTHORS: DEAR REVIEWER, THANK YOU. WE HAVE DETAILED THE INCLUSION AND EXCLUSION CRITERIA.

There is clearly 'almost nothing going on' due to the lack of a priori hypotheses.

AUTHORS: DEAR REVIEWER, THANK YOU. WE HAVE ADDED THREE CLEAR OBJECTIVES IN THIS NEW VERSION. THUS, WE HOPE THAT THE READING ARE EASIER TO FOLLOW.

RESULTS

The authors are recommended to report exclusion criteria and motives for excluding the records in the screening step.

AUTHORS: DEAR REVIEWER, THANK YOU. WE HAVE ADDED SUCH INFORMATION IN THE SECTION 3.1.

The authors split their remained 35 papers into six subsections. However, the rationale is not provided in the introduction section and, most importantly, this decision limits interpretation. How can we draft conclusions based on decision-time based on n = 3? I continue to not fully understand the hypotheses that are (or not) presented. If they want to maintain their results, information is required.

AUTHORS: DEAR REVIEWER, WE DO AGREE WITH YOU. BASED ON THAT, WE HAVE DRASTICALLY CHANGED THE RESULTS BASED ON YOUR SUGGESTION. 

The authors report “the studies varied between 0.68 and 0.86…” however they do not report cutoffs for each category of study quality. Please insert in the method section

AUTHORS: DEAR REVIEWER, THANK YOU. WE HAVE IMPROVED THIS SECTION.

Information in the Tables is extensive and confusing.  The authors are recommended to review their data display so that readers are able to understand what is going on. Additionally, the text should help readers on understanding what is going on in the respective tables.

AUTHORS: DEAR REVIEWER, WE HAVE KEPT THE TABLES BECAUSE OUR AIM WAS TO SUMMARIZE THE MAIN FINDINGS OF EACH ARTICLE.

Still in Table 1, what does “experienced” rugby players mean? Isn’t the study objective youth sport as stated in the title?

AUTHORS: DEAR REVIEWER, THANK YOU. THIS PARTICULAR CASE WAS REMOVED.

At this phase, I truly am having some problems understanding the authors' objective.

AUTHORS: DEAR REVIEWER, THANK YOU. WE HAVE ADDED THREE CLEAR OBJECTIVES IN THIS NEW VERSION. THUS, WE HOPE THAT THE READING ARE EASIER TO FOLLOW.

DISCUSSION

I will not comment much on the Discussion as I believe it will change considerably if the authors follow my advice on previous sections. Overall, I advise beginning this section by providing the study’s answer to the specific research question (from the introduction section). What did the authors intend to examine?. Then, review findings in relation to previous research and theories – are they consistent, inconsistent? If so/not… then why?. Review implications of findings for (a) existing understanding of the topic of decision-making; (b) future research, research methods; and (c) applied practice in the sport context. Please consider limitations of the study (currently no limitations are presented) and assess the impact/meaning of these limitations when trying to draw inferences/implications.

AUTHORS: DEAR REVIEWER, THANK YOU. WE HAVE DRASTICALLY CHANGED THE MANUSCRIPT AND THE DISCUSSION IS NOW FOLLOWING THE THREE OBJECTIVES AND THE THREE TOPICS OF THE RESULTS. WE HAVE ADDED ALSO A SUB-SECTION OF STUDY LIMITATIONS AND FUTURE STUDIES.

CONCLUSION

This section will arguably receive a lot of revisions after considering the reviewers' suggestions. Authors are advised to conclude by re-summarizing the findings, and their likely meaning/impact in light of those limitations and offer a clean ‘take-home-message’ that the study findings defensibly justify.

AUTHORS: DEAR REVIEWER, THANK YOU. WE HAVE DRASTICALLY CHANGED THE CONCLUSIONS.

Other comments:

With such big data, a meta-analytical approach is advised. It would increase the quality of the authors’ work and give more robust information on the proposed subject. I understand that this might seem to be a ‘hurdle’ for the authors to get done. However, there are possible issues facing current interpretations.

AUTHORS: DEAR REVIEWER, WE MAY UNDERSTAND YOUR COMMENT. HOWEVER, META-ANALYSIS IS NOT ADEQUATE IN ANY CASE, AS MENTIONED IN COCHRANE. WE DO BELIEVE THAT THE META-ANALYSIS IT IS NOT ADVISABLE TO PROCEED WITH A META-ANALYSIS IN THIS CASE. THE SMALL NUMBER OF ARTICLES, THE DIVERSITY OF METHODS AND TESTING PROCEDURES, AS WELL AS THE DIFFERENCES IN MODES OF EVALUATION AND ANALYZED JOINTS, RENDER IT IMPOSSIBLE TO PERFORM A META-ANALYSIS.

English writing in the current manuscript is not for publication and requires major improvements. There are frequent grammatical and typographical errors present throughout the paper, which mar the reader’s understanding of the research. Overall, the paper would benefit from a review from a native English speaker or proofread by editing service.

AUTHORS: DEAR REVIEWER, THANK YOU. WE HAVE SUBMITTED THE ARTICLE FOR A REVISION OF THE ENGLISH BY A NATIVE SPEAKER SPECIALIZED IN SCIENTIFIC WRITING.

Round 2

Reviewer 1 Report

INTRODUCTION

Please define clearly the difference between Experience and Expertise in sport. 

I recommend you to choose a different criteria in this case like ("difference level of ability" ..)

MATERIALS AND METHODS

237. Is it Taylor & Francis a database or an editorial?

241-242. In the inclusion criteria... how can be a young player an expert player?

RESULTS

In PRISMA flow diagram...  which are the reasons for excluding 5018 records?

In the last diagram... there are no other papers about the topic published in other languages to exclude?

(41, 42, 43) How could be a 7,9, 10 years old player an expert?? 

(43) This paper could be in age differences because of the sample? 

Table 2  title is the same of table 1.

Study (50) could be in table 1? There is no different groups of age..

DISCUSSION

572-573. That´s the key. May be most of the sample of the study are just players with different levels of ability, skills or experience (not expertise). How could you define talent in a 7 years old player? 

4.3. The authors consider video as a training program and there are other studies in table 1 and 2 which include video in the design.

CONCLUSIONS

Expert and novice players are the same than talented or not talented players? 

Author Response

INTRODUCTION

Please define clearly the difference between Experience and Expertise in sport. 

I recommend you to choose a different criteria in this case like ("difference level of ability" ..)

AUTHORS: The authors would like to thank you the Reviewer’s comments. To better clarify, it was changed in “novices and expert players” to “young players with different level of ability”.

MATERIALS AND METHODS

  1. Is it Taylor & Francis a database or an editorial?

AUTHORS: Dear Review, thank you for this comment. In fact, for this search it was used Taylor & Francis online database. It was clarified in the manuscript.

241-242. In the inclusion criteria... how can be a young player an expert player?

AUTHORS: Dear Reviewer, thank you for your comment. We changed the this the terminology in the text, however, it was used based on the studies included in this review.

RESULTS

In PRISMA flow diagram...  which are the reasons for excluding 5018 records?

AUTHORS: Dear Reviewer, thank you for your comment and to give the chance to clarify that elimination. In fact, most of the studies excluded included analysis of decision-making outside the sports field (mostly economics and psychology). This information was added in the manuscript.

In the last diagram... there are no other papers about the topic published in other languages to exclude?

AUTHORS: Dear Reviewer, in fact, at this stage we have already excluded some other studies in different languages, this Spanish study had the abstract in English, that is why it stayed for the last stage of selection.

(41, 42, 43) How could be a 7, 9, 10 years old player an expert?? 

AUTHORS: Dear Reviewer, thank you for highlighting that. Although we used the expressions applied on the included studies, the authors agree with the Reviewer. Therefore, it was changed along the manuscript.

(43) This paper could be in age differences because of the sample? 

AUTHORS: Dear reviewer, thank you. We do agree with you. We have added in the table 2 also.

Table 2 title is the same of table 1.

AUTHORS: Dear Reviewer, thank you for highlight that. The title was changed.

Study (50) could be in table 1? There is no different groups of age.

AUTHORS: Dear reviewer, thank you. We have updated the sample information because the study of González-Víllora tested different age groups (14 players with less than 8 yo; 13 players with less than 10 yo; 14 players with less than 12 yo; and 16 players with less than 14 yo). However he did not tested different skill abilities and for that reason we will keep in table 2.

DISCUSSION

572-573. That´s the key. May be most of the sample of the study are just players with different levels of ability, skills or experience (not expertise). How could you define talent in a 7 years old player? 

AUTHORS: Thank you for your comment, the author agree with the Reviewer.

4.3. The authors consider video as a training program and there are other studies in table 1 and 2 which include video in the design.

AUTHORS: Dear reviewer, thank you. However, training program consisted not in video-analysis to test decisions but only in groups that had intervention to develop the decision-making based on video sessions.

CONCLUSIONS

Expert and novice players are the same than talented or not talented players? 

AUTHORS: Dear Reviewer, thank you for your comment. We decided to remove that words to avoid confusion.

Reviewer 2 Report

thank you for asking me to review the manuscript ID 771551: Decision making in youth sport: a systematic review. The authors improved greatly their work. However, there are still some concerns that need to be addressed, which I list below.

INTRODUCTION

The introduction section was greatly revised.

Some small comments to consider:

Page 3, line 117-121. I think the readers would like to know the conclusions of the narrative and systematic reviews reported by the authors. Are there some inconsistencies among them? Or are some associations that could lead to an avenue of more systematic approaches?

Provide some hypotheses based on your literature review.

METHODS

Please report the exact time of study search (from when till when).

Please justify the selection of databases.

I still do not understand the “maturation” definition. Additionally, the terms “expert” and “maturation” and “talent” are used interchangeably throughout the paper. Are these the same or not? Please provide a definition for each type of athlete characteristic.

The authors claim that they have not received any feedback from the NIHR due to their registration of the systematic review in PROSPERO. Which is weird since they have been accepting new registrations. I will not push this; however, I recommend the authors to report PROSPERO registration number after publication.

RESULTS

The authors continue to not explain why 5018 records were excluded in the “screening” step. Please provide the motives for excluding them.

Page 6, line 212-216. The authors continue to report explain the minimum and maximal values. How where they calculated? What are the cutoffs? And do you interpret them?

The authors’ decision to revise their sub-topics in an asset. It is clearer for readers to understand what is going on and how can we examine the results.

I suggest the authors to provide an overall summary of the k studies. That is, how many participants encompasses this systematic review, as well as their characteristics (e.g., gender, country).

DISCUSSION

I advise the authors on beginning this section by providing the study’s answer to the specific research question (from the introduction section) or what was intended to do.

Then discuss the results based on the three proposed tables by the authors.

CONCLUSION

The conclusion improved greatly. However, I still insist on giving a take-home message. What will researchers, coaches, and other professionals retrieve from your work?

Other comments:

The authors report making substantial revisions throughout the entirety of their study, including a review by a “native speaker specialized in scientific writing”. The English of the manuscript is better in this iteration. Still, there remain a number of issues of word choice, comma placement, and other nuances that should be improved.

Author Response

INTRODUCTION

The introduction section was greatly revised.

AUTHORS: The authors acknowledge the reviewer’s words

Some small comments to consider:

Page 3, line 117-121. I think the readers would like to know the conclusions of the narrative and systematic reviews reported by the authors. Are there some inconsistencies among them? Or are some associations that could lead to an avenue of more systematic approaches?

AUTHORS: Dear reviewer, thank you. We have tried to detail.

Provide some hypotheses based on your literature review.

AUTHORS: Dear Reviewer, thank you for your comment. Hypothesis were included at the end of the introduction session.

METHODS

Please report the exact time of study search (from when till when).

AUTHORS: Dear Reviewer, thank you for your comment. The exact time was added in the text.

Please justify the selection of databases.

AUTHORS: Dear Reviewer, thank you for your comment. A clear justification was added on the text.

I still do not understand the “maturation” definition. Additionally, the terms “expert” and “maturation” and “talent” are used interchangeably throughout the paper. Are these the same or not? Please provide a definition for each type of athlete characteristic.

AUTHORS: Dear Reviewer, thank you for your comment. Considering your comments and the comments of Reviewer 2, the authors changed the words “expert/ non expert” and “talented/less talented” to “low and high level of ability” as we are analyzing young athletes.

The authors claim that they have not received any feedback from the NIHR due to their registration of the systematic review in PROSPERO. Which is weird since they have been accepting new registrations. I will not push this; however, I recommend the authors to report PROSPERO registration number after publication.

AUTHORS: Dear reviewer, thank you. Actually, we had no feedback at time, however we also registered our protocol at the International Platform of Registered Systematic Review and Meta-analysis Protocols under number 202040207 and DOI 10.37766/inplasy2020.4.0207.

RESULTS

The authors continue to not explain why 5018 records were excluded in the “screening” step. Please provide the motives for excluding them.

AUTHORS: Dear Reviewer, thank you for your comment. Those papers were excluded mainly because they provided information about decision-making in other fields as economics and psychology. Those justifications were included in the text.

Page 6, line 212-216. The authors continue to report explain the minimum and maximal values. How where they calculated? What are the cutoffs? And do you interpret them?

AUTHORS: Dear Reviewer, thank you for your comment. The ages reported in this section were related to the studies included. For this review, the authors only included young athletes (up to 18 years old), here we only reporting what the studies included.

The authors’ decision to revise their sub-topics in an asset. It is clearer for readers to understand what is going on and how can we examine the results.

AUTHORS: Dear Reviewer, thank you for your kind words. The authors made an effort to better clarify the results regarding the Reviewers suggestions.

I suggest the authors to provide an overall summary of the k studies. That is, how many participants encompasses this systematic review, as well as their characteristics (e.g., gender, country).

AUTHORS: Dear reviewer, thank you. We have made a sum of participants per sex and add as description in each result section.

DISCUSSION

I advise the authors on beginning this section by providing the study’s answer to the specific research question (from the introduction section) or what was intended to do.

Then discuss the results based on the three proposed tables by the authors.

AUTHORS: Dear Reviewer, thank you for your comment. An introductory paragraph was added to the text as suggested.

CONCLUSION

The conclusion improved greatly. However, I still insist on giving a take-home message. What will researchers, coaches, and other professionals retrieve from your work?

AUTHORS: Dear reviewer, thank you. We have added an additional paragraph in conclusions about practical implications.

Other comments:

The authors report making substantial revisions throughout the entirety of their study, including a review by a “native speaker specialized in scientific writing”. The English of the manuscript is better in this iteration. Still, there remain a number of issues of word choice, comma placement, and other nuances that should be improved.

AUTHORS: Dear reviewer, thank you. We have tried to improve the writing.

Round 3

Reviewer 1 Report

I appreciate your effort.  The last version of the manuscript has improved considerably.

However still need to modify some aspects:

  • I recommend the authors to rewrite the title of the different tables in relation to the content, doing them more specific. Ex: Table 1:  participants (level of performance/ability).  Same in others... Design in table 3 could be renamed Intervention or training program. Also table 3 should be completed with control and experiment group and pre- and post.

Furthermore, I recommend the authors to add a table resume including  the different sports separated, just to know how is the state in each of them about this topic (decision making).

This study Schorer et al. [43] doesnt match with the criteria. (Adults and senior )   

Any differences between men and women  should be remarked? 

There is a lack of references which could match with the aims of this study. I should recommend the authors to review and update the search.

Example:

Fortes, L. S.; Freitas-Júnior, C. G.; Paes, P. P.; Vieira, L. F.; Nascimento-Júnior, J. R.; Lima-Júnior, D. R. A.A.; Ferreira, M. E. Effect of an eight-week imagery training programme on passing decision-making of young volleyball players.

And others... 

Author Response

I appreciate your effort.  The last version of the manuscript has improved considerably.

AUTHORS: DEAR REVIEWER, THANK YOU. WE HAVE TRIED BASED ON YOUR COMMENTS AND SUGGESTIONS.

However still need to modify some aspects:

  • I recommend the authors to rewrite the title of the different tables in relation to the content, doing them more specific. Ex: Table 1:  participants (level of performance/ability).  Same in others... Design in table 3 could be renamed Intervention or training program. Also table 3 should be completed with control and experiment group and pre- and post.

AUTHORS: DEAR REVIEWER, THANK YOU. WE HAVE CHANGED THE TABLES, NAMELY ADDING MORE SPECIFIC TITLE BASED ON YOUR RECOMMENDATIONS. IN TABLE 3 (NOW TABLE 4) WE HAVE ADDED INFORMATION ABOUT CONTROL AND INTERVENTION GROUPS AND PRE-POST MEASURES

Furthermore, I recommend the authors to add a table resume including the different sports separated, just to know how is the state in each of them about this topic (decision making).

AUTHORS: DEAR REVIEWER, THANK YOU. WE HAVE ADDED A NEW TABLE (Nº.1) WITH INDICATION OF STUDY, TEAM SPORT AND GENDER.

This study Schorer et al. [43] doesnt match with the criteria. (Adults and senior )

AUTHORS: DEAR REVIEWER, THANK YOU. ACTUALLY, THE STUDY OF SCHORER HAD PLAYERS FROM 14.4 TO 46.7 YO, THUS BEING INCLUDED BASED ON THE FACT THAT TWO GROUPS (SUB-YOUTH AND YOUTH) WERE UNDER -18 PLAYERS.

Any differences between men and women should be remarked? 

AUTHORS: DEAR REVIEWER, THANK YOU FOR YOUR COMMENT. IN FACT, ALTHOUGH SIX STUDIES INCLUDED FEMALE AND MALE PLAYERS, ONLY TWO OF THEM COMPARED SEXES. THE OTHER FOUR STUDIES JUST INCLUDE BOTH SEXES TOGETHER IN THE SAMPLE, PRECLUDING COMPARISIONS. NEVERTHELESS, THE RESULTS OF THOSE TWO STUDIES WERE DISCUSSED AND INCLUDED IN THE DISCUSSION SESSION.

There is a lack of references which could match with the aims of this study. I should recommend the authors to review and update the search.

Example:

Fortes, L. S.; Freitas-Júnior, C. G.; Paes, P. P.; Vieira, L. F.; Nascimento-Júnior, J. R.; Lima-Júnior, D. R. A.A.; Ferreira, M. E. Effect of an eight-week imagery training programme on passing decision-making of young volleyball players.

And others... 

AUTHORS: DEAR REVIEWER, THANK YOU. WE HAVE INCLUDED THE ARTICLE AND DISCUSSED THE FINDINGS.

Reviewer 2 Report

The introduction section seems to be suitable for this study.

The authors state registration number which provides more confidence to the reader.

Quick grammar and spelling checks are still needed. Please consider this during proofreading.

Author Response

The introduction section seems to be suitable for this study.

AUTHORS: DEAR REVIEWER, THANK YOU.

The authors state registration number which provides more confidence to the reader.

AUTHORS: DEAR REVIEWER, THANK YOU.

Quick grammar and spelling checks are still needed. Please consider this during proofreading.

AUTHORS: DEAR REVIEWER THANK YOU. WE HAVE TRIED TO IMPROVE DURING THIS ROUND.